# Oxidative Stress and Cataract Formation: Evaluating the Efficacy of Antioxidant Therapies

**DOI:** 10.3390/biom14091055

**Published:** 2024-08-25

**Authors:** Merve Kulbay, Kevin Y. Wu, Gurleen K. Nirwal, Paul Bélanger, Simon D. Tran

**Affiliations:** 1Department of Ophthalmology & Visual Sciences, McGill University, Montreal, QC H4A 3S5, Canada; merve.kulbay@mail.mcgill.ca; 2Division of Ophthalmology, Department of Surgery, University of Sherbrooke, Sherbrooke, QC J4K 0A8, Canada; yang.wu@usherbrooke.ca (K.Y.W.);; 3Department of Zoology, University of British Columbia, Vancouver, BC V6T 1Z4, Canada; 4Faculty of Dental Medicine and Oral Health Sciences, McGill University, Montreal, QC H3A 1G1, Canada

**Keywords:** cataract formation, oxidative stress, antioxidants, treatments

## Abstract

This comprehensive review investigates the pivotal role of reactive oxygen species (ROS) in cataract formation and evaluates the potential of antioxidant therapies in mitigating this ocular condition. By elucidating the mechanisms of oxidative stress, the article examines how ROS contribute to the deterioration of lens proteins and lipids, leading to the characteristic aggregation, cross-linking, and light scattering observed in cataracts. The review provides a thorough assessment of various antioxidant strategies aimed at preventing and managing cataracts, such as dietary antioxidants (i.e., vitamins C and E, lutein, and zeaxanthin), as well as pharmacological agents with antioxidative properties. Furthermore, the article explores innovative therapeutic approaches, including gene therapy and nanotechnology-based delivery systems, designed to bolster antioxidant defenses in ocular tissues. Concluding with a critical analysis of current research, the review offers evidence-based recommendations for optimizing antioxidant therapies. The current literature on the use of antioxidant therapies to prevent cataract formation is sparse. There is a lack of evidence-based conclusions; further clinical studies are needed to endorse the use of antioxidant strategies in patients to prevent cataractogenesis. However, personalized treatment plans considering individual patient factors and disease stages can be applied. This article serves as a valuable resource, providing insights into the potential of antioxidants to alleviate the burden of cataracts.

## 1. Introduction

The anterior segment of the eye encompasses various ocular pathologies, with cataracts being the most common. Cataracts is amongst the leading causes of blindness globally [1]. Cataracts are characterized by the gradual opacification of the normally clear lens of the eye, which has as consequence to alter light refraction, therefore leading to blindness if left untreated. In the last 30 years, the prevalence of moderate to severe vision impairment due to cataract was shown to have increased by nearly 30% [1]. The burden associated with cataracts is significant; the disability-adjusted life years (DALY), which represents the sum of the years of life lost due to premature mortality and the years lived with a disability (World Health Organization), was shown to have rose by 91.2% from 1990 to 2019 [2]. Furthermore, the global economic burden of cataract was reported to reach as high as 3654 USD ppp [3]. To leverage these socioeconomic impacts worldwide, significant advances have been made in the surgical management of cataracts over the years. However, disparities in access to an ophthalmologist for subsequent management are present in populations with lower socioeconomic status, as well as some ethnic groups [4,5,6].

Cataract formation can be influenced by various factors, with the most common cause being age-related modifications [7]. Lens opacification can be induced by traumatic injuries (e.g., perforating or blunt traumas, electric shock, ultraviolet or ionizing radiations, and chemical injuries), systemic diseases (e.g., neurofibromatosis type 2, myotonic dystrophy), endocrine diseases (e.g., diabetes mellitus), primary ocular diseases, drugs, poor nutrition, alcohol use disorder, and smoking [7].

Reactive oxygen species (ROS) are the backbone of cataract formation, and their importance and central role in cataract formation has been thoroughly investigated in the past years [8,9,10]. Depletion of antioxidants and subsequent accumulation of ROS contribute to cataractogenesis, through the alternation of cell signaling pathways involved in cell growth and death. To better understand the novel strategies in addressing cataract formation, this review presents a thorough summary of lens physiology ensued by cataractogenesis pathogenesis, while providing an emphasis on the pivotal role of ROS.

Currently, no clear guidelines exist for the use of antioxidants in the treatment or prevention of cataract formation. To our best knowledge, the importance of these agents for the management of cataracts is yet to be discussed critically. Therefore, this comprehensive literature review aims to discuss for the first time the limitations of antioxidative treatments and the future perspectives from the standpoint of clinicians.

## 2. The Structure and Physiology of the Lens

The refractive index represents the backbone of vision and is a defining characteristic of an optic material. As the light goes through its path from the anterior segment to the posterior segment of the eye, it is reflected and refracted through the structural components it encounters, such as the cornea and the lens (Figure 1). Alterations to the structure of these components, as well as to other constituents such as the vitreous humor, can alter the stability and the refractive index of each respective structure. Disturbances in the refractive index in the lens forms the fundamental principles of vision loss in cataractogenesis. In this section, we will review the physiology and the regulatory mechanisms present in the human lens against endogenous and exogenous stress, followed by the pathophysiology of cataract formation, with an emphasis on the role of ROS.

### Physiology of the Lens and Its Redox Regulatory Mechanisms

The lens of the eye constantly faces oxidative stress from endogenous (e.g., cellular metabolic activities and inflammation) and exogenous (e.g., ultraviolet, ionizing, and gamma radiations, as well as X-rays) sources. Chronic exposure to these biohazards leads to lens opacification (i.e., cataract formation) and subsequent loss of the high refractive index of the lens.

The human lens is a biconvex transparent structure and is derived from the surface ectoderm. It consists of four distinct parts: the lens zonules, capsule, epithelium, and fibers, from the lateral to the medial aspect (Figure 2). The lens capsule is a thin and transparent layer that is mainly composed of type IV collagen and laminin networks [11]. It has elastic properties, but lacks any elastic tissue. The epithelial cell layers can be divided into three zones based on their function: the central zone, intermediate zone, and germinative zone [12]. The central zone, which encompasses epithelial cells in the anterior lens capsule, involves cuboidal cells with round apically located nuclei and characterized by the absence of mitosis. These cells decline in number with age. As they progress through the intermediate zone, the epithelial cells become smaller and cylindrical and start undergoing mitosis. In the germinative zone, the epithelial cells are mostly columnar, located pre-equatorially, and are actively dividing to form the lens fibers. As they divide, the epithelial cells migrate posteriorly to form the lens fibers. They are further arranged in a specific 3D orientation to form the nucleus and the cortex of the lens. Finally, the lens zonules, which have a central role in accommodation, are composed of cysteine and they regress with age [13]. Clinically, cataract subtypes can be classified according to an abnormality in these processes. For example, anterior subcapsular cataracts are characterized by metaplasia occurring in the central zone, whereas posterior capsule opacification is the result of the migration of equatorial cells towards the posterior capsule.

The high refractive index of the lens is achieved through its transparency, which further depends on the spatial arrangement of the lens fibers and the solubility of lens proteins (i.e., crystallins) [14]. In the lens, 3 types of crystallins exist: α-, β-, and γ-crystallins. These proteins are subjected to post-translational modifications [15], which are the target of exogenous and endogenous stress sources. The α-crystallin is the major constituent of the lens and is a member of the small heat shock protein (sHSP) family [16,17]. In addition to proving structural support to the lens, which is crucial in maintaining the refractive index of the lens, α-crystallin acts as in a chaperone-like manner by suppressing thermally induced protein aggregation: the binding of α-crystallin with non-native proteins prevents aggregation and maintains the substrates in an adequately folded structure [16]. The ubiquitin–proteasome pathway in lens fibers allows adequate discrimination between normal and damaged proteins, further preventing cataract formation [18]. The α-crystallin has two isoforms: the αA- and αB-crystallins. Although β- and γ-crystallins are not members of the sHSP family, they have been shown to exhibit similar structural roles, where their presence is important for lens stability, transparency, and subsequent refractive index [19].

Lens opacification occurs at three distinct regions and can be used to classify cataracts; lens opacification leads to nuclear, posterior subcapsular, cortical, or mixed cataracts. However, multiple endogenous antioxidative mechanisms exist in the human lens to prevent excessive ROS production and subsequent opacification at early stages (Figure 3). The first line of defense against oxidative stress in the lens involves the presence of an extensive network of gap junctions. These junctions allow the passage of antioxidant molecules (up to 1 kDa) within the human lens, as well as facilitating intercellular communication [20,21]. Gap junctions in the lens are composed of specific channels, known as connexins. Three isoforms are expressed within the lens: connexins-43 (Cx43), -46 (Cx46), and -50 (Cx50) [22]. Connexin channels have shown to be involved in lens microcirculation regulation through hydrostatic pressure maintenance in the context of high fluid flow shear stress [23], as well as glucose and antioxidant delivery [24], processes that will be further discussed. Lens antioxidant systems are under the regulation of nuclear factor-erythroid 2-related factor 2 (Nrf2). Nrf2 is a transcription factor that controls the expression of antioxidant enzymes (i.e., glutathione (GSH), superoxide dismustase (SOD), catalase), as well as HSP [25,26]. Under homeostatic conditions in the human lens, Nrf2 is bound to its negative regulator, kelch-like ECH-associated protein 1 (Keap1), therefore maintaining low levels of Nrf2 through the ubiquitin-proteasome pathway [27]. Conversely, in a pro-oxidative environment, Nrf2 dissociates from Keap1 to translocate to the nucleus and bind the antioxidant response element (ARE) in the promoter region, to activate the transcription of antioxidant enzymes [14,28]. These regulatory mechanisms are highly influenced by the endoplasmic reticulum (ER) stress pathway [28]. Ma and colleagues have shown that H_2_O_2_-induced oxidative stress induces the activation of Nrf2 and ATF4 (i.e., activating transcription factor 4), with subsequent facilitated induction of the expression of ARE-dependent antioxidant enzyme [28]. The activation the ER stress pathways prevents the accumulation of misfolded proteins through the induction of the unfolded protein response (UPR) pathway. The UPR pathway is induced by three distinct signaling pathways, which are the protein kinase RNA (PKR)-like endoplasmic reticulum kinase (PERK), the inositol requiring kinase 1 (IRE1), and the activating transcription factor 6 (ATF6) pathways [29]. The UPR pathway orchestrates the activation of numerous effectors that ultimately lead to the nuclear transcription of enzymes involved in protein folding and degradation of misfolded proteins. Overall, dysregulations in the antioxidative systems in the human lens can lead to cataract formation.

The antioxidative systems in the human lens can be divided into two categories: the enzymatic and the non-enzymatic antioxidants. Enzymatic antioxidants in the lens encompass SOD, catalase, and glutathione peroxidase (GPX). SOD catalyzes the dismutation of superoxide molecules into oxygen molecules [30]. SOD is known to prevent the apoptosis of lens epithelial cells (LECs) [31]. Catalase, which is involved in processing hydrogen peroxide (H_2_O_2_) to water and oxygen molecules [32], plays a key role in preventing cataract formation through the suppression of transforming growth factor-B (TGF-B); catalase was shown to suppress LEC surface blebbing and apoptosis [33]. GPX, which has a similar function to catalase, is also a crucial antioxidant enzyme in the human lens [34]. Loss of GPX expression through gene editing in mice was shown to lead to accelerated age-related cataract formation [35]. Non-enzymatic antioxidants, such as GSH, vitamin C, and vitamin E, also play a key role in redox balance. The role of GSH in the prevention of lens opacification has been thoroughly reported [36,37,38]. GSH is highly expressed within the human lens and is involved in the detoxification of numerous damaging oxidants [39]. Exogenous drugs involved in the upregulation of GSH levels were shown to prevent cell death of LECs [40]. Finally, vitamin C and Vitamin E were shown to reduce oxidative stress in LECs [41]. Genetic factors and environmental factors can alter the functionality of the antioxidant systems within the LECs and therefore accelerate cataract formation. A study conducted in 2000 by Hammond et al. aimed to outline the genetic and environmental factors involved in nuclear cataracts by evaluating monozygotic and dizygotic twins [42]. Genetic factors were shown to account for almost 50% of the observed variations in disease severity, whereas environmental factors accounted for almost 15% [42]. Amongst the genetic variations, a more recent study from Zhu et al. demonstrated that DNA hypermethylation downregulates the expression of certain antioxidant genes (i.e., *GSTP1* and *TXNRD2*) in LECs and therefore leads to early cataract formation [43]. Conversely, exposure to ultraviolet radiation was shown to accelerate apoptosis of LECs [44]. A retrospective study in Iceland demonstrated that participants with greater outdoor exposure was associated with increased risk of cortical cataract formation [45]. Although genetics factors cannot be controlled, one could limit their exposure to environment factors with the aim of reducing the rate of cataract formation. However, these should not be strongly recommended to patients, given the lack of evidence-based data, which is discussed in ensuing sections.

## 3. Ocular Damages Induced by Reactive Oxygen Species: Cataracto-genesis

The pivotal role of oxidative stress in the pathogenesis of numerous ocular pathologies has been thoroughly discussed and reviewed [46,47,48]. Defects in antioxidative systems in the human lens leading to subsequent protein aggregation are the mainstay of lens opacification and cataract formation. A recent retrospective study has shown that the aqueous humor of patients with high grade cataract exhibited lower total antioxidant status compared to those with lower grades [49]. However, no significant changes were observed according to cataract subtype (i.e., nuclear, posterior subcapsular, and mixed). In this section, we review the main hallmarks of ROS-mediated cataract formation, with a focus on most recent advances.

### 3.1. Antioxidative Systems

As outlined earlier, connexins are key players in the maintenance of redox homeostasis. In lens epithelial cells, H_2_O_2_- and UVB-induced oxidative stress was shown to activate Cx43 hemichannels (HCs) and promote an antioxidative environment through GSH synthesis and exogenous uptake [50]. Fluid flow shear stress was similarly shown to activate Cx43 HCs and promote subsequent GSH release [51]. Released GSH by Cx43 HCs was further shown to be taken by lens fiber Cx50 HC; an inhibition of Cx50 HCs increases H_2_O_2_-induced apoptosis in differentiated lens epithelium cells [51]. Reduced GSH possesses great antioxidant capacity by preventing the accumulation of ROS; topical GSH significantly decreased cataract formation in a rat model of sodium selenite-induced cataract [52]. Furthermore, haploinsufficiency of Cx43 in mouse lens was shown to promote an oxidative stress environment which favored cataract formation [50]. In heterozygous Cx43-null (Cx43^+/−^) mice lens, Quan and colleagues have demonstrated decreased levels of SOD1 and GPX1 in the anterior and equatorial epithelial regions [50]. These studies outline the importance of Cx HCs in the maintenance of an antioxidative microenvironment in the human lens.

### 3.2. Protein Aggregation, Cross-Linking, and Light-Scattering

A current well-known mechanism involved in cataract formation involves the formation of light-scattering aggregates by damaged crystallins [53]. Light scattering α- [54,55], β- [56,57], and γ-crystallin [58] aggregate formation in cataract formations has been thoroughly demonstrated within the past years. Human lenses lack the ability of protein turnover [59]. Therefore, α-crystallins encompass a crucial molecular chaperone function to prevent protein misfolding and subsequent aggregate accumulation. Oxidation, deamination, and tryptophan derivatization are well-known post-translational modifications that have been shown to be involved in age-related cataracts [60,61,62]. Formation of disulfide bonds, liquid–liquid phase separations, and domain swapping (i.e., the process by which identical monomeric proteins swap structural elements to induce oligomers) induce opacities of the lens [63,64]. UV radiations induce the production of ROS, which have been shown to oxidize α-crystallin amino acids, such as methionine, tryptophan, and cysteine residues, and lead to cross-linking and oligomerization [65,66]. More recently, it was shown that low concentrations of Cu(II) in α-crystallins promoted its chaperone activity and further inhibited its aggregation by maintaining surface stability [55]. With aging, deamination of γ-crystallins was shown to occur and induce significant destabilization of the crystallin [67]. Vetter and colleagues have investigated the process by which γ-crystallin aggregates induce cataract formation [68]. By forming a γS deaminated crystallin through aspartate residue mutation, it was shown that deamination increased the susceptibility of the crystallin to oxidation, and increased protein unfolding and aggregate formation [68]. Conversely, exogenous ATP was shown to inhibit the γS-crystallin aggregate formation, suggesting that an age-related ATP depletion could be linked to cataracto-genesis [69]. Interactions of crystallins, mainly αB-crystallins, with the human lens lipid membranes through hydrophobic interaction are also involved in cataractogenesis [70]. These interactions were shown to be regulated by the composition in cholesterol and cholesterol bilayer domains (CBDs): higher cholesterol and CBD content decreases the membrane hydrophobicity, therefore decreasing crystallin binding [71].

Furthermore, numerous mutations in crystallins have been shown to facilitate cataract formation. In γS-crystallins, the S39C variant was shown to enhance protein sensitivity to environmental triggers and expediate crystallin denaturation induced by heat and UV light [72]. Similar results were obtained in βA3-crystallin isoform with the R114C variant [73]. The D109H and R69C variants of αB-crystallins was shown to be involved in the pathogenesis of congenital cataracts [74]. Overall, cataractogenesis is a complex pathway that involves primarily an oxidative environment, which further leads to dysregulations in antioxidant systems and the ER stress pathway.

### 3.3. Lipid Peroxidation and Loss of Membrane Integrity

For many years, lipid peroxidation has been known to be a key factor in cataracto-genesis through the loss of membrane integrity [75]. Lipid peroxidation is a self-sustained process characterized by free radical oxidation of polyunsaturated fatty acids (PUFAs) (e.g., linoleic acid or arachidonic acid), therefore leading to the production of ROS [76]. Lipid peroxidation within the lipidic bilayer cell membrane was shown to impair lipid-based interactions within the lenticular fibre membranes. Patients with senile and complicated cataracts were shown to have increased concentrations of lipid peroxidation end products, such as diene conjugates, lipid hydroperoxides, and oxy-derivates of phospholipid fatty acids within their aqueous humour [77]. More recently, Kreuzer et al. demonstrated that LECs within cortical cataracts underwent lipid peroxidation, therefore leading to lens clouding, whereas nuclear cataracts exhibited greater protein aggregation in the form of fibrils [78]. This mechanism was shown to increase the susceptibility of LECs to ferroptosis; glutathione peroxidase inhibitors (i.e., RSL3) in human LECs in vitro was shown to induce the accumulation of intracellular redox-active iron through the downregulation of the iron exporter ferroportin gene (i.e., *SLC40A1*) [79,80]. Pretreatment of human LECs in vitro with potent antioxidant compounds, such as N-acetylcysteine amide (NACA), N-acetylcysteine (NAC), and GSH, showed greater cell viability following *tert*-butyl hydroperoxide-induced lipid peroxidation [40]. NACA was shown to be of greater efficiency in preventing lipid peroxidation-mediated damages through the upregulation of GSH concentrations and subsequent decrease in ROS [40]. Aldose reductase, a NADPH-dependent oxidoreductase, was further shown to play a key role in LECs in regard to cellular detoxification following 4-hydroxy-2-nonenal-induced lipid peroxidation [81]. Similarly, keratinocyte growth factor-2 (KGF-2) was shown to exhibit cytoprotective effects against H_2_O_2_-induced damages in human LECs and rat lenses; KGF-2 was shown to increase the expression of B-cell lymphoma-2 (Bcl-2), SOD2, and catalase, while decreasing the expression of cleaved caspase-3 and Bcl-2-associated X (Bax) [82]. Overall, these results suggest a key role of lipid peroxidation in cataractogenesis through the regulation of LEC function and integrity. The engineering of novel strategies that target lipid peroxidation shows a promising avenue in preventing cataractogenesis. A possible approach would be to enhance antioxidant levels within LECs through the upregulation of antioxidant enzymes.

## 4. Antioxidant Strategies for the Prevention and Management of Cataracts

### 4.1. Dietary Nutrients and Supplements

#### 4.1.1. Vitamins C and E

Vitamins C and E are antioxidants in the human lens. Vitamin C endogenously scavenges and quenches ROS and free radicals [83]. Sources of vitamin C include citrus fruits, potatoes, and tomatoes [84]. It is also able to absorb UV light, preventing oxidative damage [85]. Vitamin E attenuates ROS production during fat oxidation and free radical reaction propagation [86]. Dietary sources of vitamin E include seeds, nuts, and green leafy vegetables, among other sources [86].

Preclinical studies have demonstrated the potential benefit of vitamin C and E consumption for cataract prevention and attenuation. In senescence marker protein-30 knockout mice (incapable of synthesizing vitamin C) with UVR-B-induced cataracts, access to water with 1.5 g/L of vitamin C was demonstrated to cause less extensive lens opacity than in knockout mice without access to vitamin C [87]. It is thus evidenced that vitamin C depletion can make the lens more susceptible to cataracts. Similarly, in a study testing supplementation with 50 mg/kg/day of vitamin E in female rats with naphthalene-induced cataracts, values of lens glutathione, soluble protein, and water content profiles indicated an anti-cataract effect [88]. Additionally, epidemiological studies have inversely linked vitamin C and E consumption with cataracts. In a population-based study, for example, the reported use of multivitamins or supplements containing vitamins A or E decreased the 5-year risk for nuclear and cortical cataracts [89]. A prospective comparative study found lower blood levels of vitamin E in nuclear and cortical cataract patients [90]. However, while animal and epidemiological studies may suggest a role for these vitamins in cataract prevention, there is a lack of concrete clinical evidence to support an appreciable effect of their consumption [91].

A large-scale, double-masked, placebo-controlled trial spanning 8 years noted that alternate-day consumption of vitamin E and/or daily consumption of vitamin C did not have a significant effect on cataract incidence [92]. Similarly, a study conducted with participants from South India found no significant difference in cataract progression between a group who received a 5-year course of β carotene, vitamin C, and vitamin E versus a placebo [93]. In a follow-up of this trial, it was also noted that there was no significant difference in number of cataract surgeries between the two groups 15 years later [94]. Other studies have also shown a lack of significant effect of vitamins C and E in cataract prevention [92,95]. Interestingly, results from a population-based prospective cohort study suggested a possible implication of high vitamin C or E doses in increasing the risk of cataracts [96].

The lack of concrete experimental data in human trials reflecting the preventative effects of vitamins C and E may discredit their role as an antioxidant strategy for preventing cataracts. While preclinical and epidemiological studies do suggest a potential application for vitamin C and E supplementation in cataract prevention, the discordance between results from these studies and those of clinical studies introduces a lack of certainty in its clinical implementation. Findings from epidemiological studies alone cannot be utilized to make conclusive statements about the impact of vitamin C and E supplementation on cataract formation. In assessing the link between vitamin C and E supplement consumption and cataract risk, unmeasured lifestyle differences may account for the identified differences in epidemiological studies assessing cataract risk as opposed to the consumption of these vitamins alone. As such, further research is necessary to resolve the lack of cohesion in the current literature concerning the benefit or lack thereof of vitamin C and E consumption to allow for evidence-based recommendations for antioxidant cataract prevention.

#### 4.1.2. Lutein and Zeaxanthin

Lutein and zeaxanthin are dietary xanthophylls that are components of the macular pigment [97]. Oxidative damage initiated by light penetration into the eye is a significant contributing factor to cataractogenesis. UV irradiation can cause photochemical generation of ROS, which leads to oxidative damage [98]. However, the peak absorption spectra of lutein and zeaxanthin help to attenuate this damage by filtering out blue and UV light [97]. Additionally, they scavenge singlet oxygens and free radicals, thus providing further antioxidative benefits [99]. Lutein can be found in leafy green vegetables, chicken egg yolks, and green peppers, whereas zeaxanthin-rich foods include corn tortillas, chicken egg yolks, and red pepper [100].

Epidemiologically, studies have shown an inverse relationship between dietary lutein and zeaxanthin intake and the occurrence of nuclear cataracts [101,102]. Meta-analyses have indicated that high blood levels of lutein and zeaxanthin have a significant association with lower nuclear cataract risk [103]. However, similar to that of vitamins C and E, there is a lack of clinical research reflecting a concrete effect of dietary lutein and zeaxanthin on cataract prevention. In a multicenter, double-masked clinical trial, AREDS2 researchers found that daily consumption of supplementary lutein and zeaxanthin did not have a statistically significant effect on the rate of vision loss and cataract surgery in participants [104]. Overall, however, due to the large prevalence of cataracts in elderly populations, even small effects of lutein and zeaxanthin on cataracts can have a strong benefit from a population health perspective. Therefore, more controlled clinical trials testing the effect of lutein and zeaxanthin supplementation on cataract prevention must be conducted to consolidate current evidence.

### 4.2. Potential Pharmacological Agents with Antioxidative Properties for Cataract Prevention and Treatment

Currently, surgical removal of cataracts and artificial lens replacement is the most effective option for cataract treatment. However, surgery can be costly for patients in developing countries. Additionally, cataract surgery can be accompanied by various complications, such as posterior capsule rupture, endophthalmitis, and the need for additional surgery [105].

Therefore, the development of pharmacological agents can help to address these concerns and improve outcomes for cataract patients. ROSs are pathogenic factors in most cataract types such as radiation cataracts, age-related cataracts, and posterior capsule opacification [106], so the investigation of antioxidant therapies is essential to this development. This section discusses antioxidative pharmacological agents that have shown potential for cataract treatment in cataract models in vitro and in vivo (Table 1).

#### 4.2.1. N-acetyl-carnosine

N-acetyl-carnosine (NACS) is a prodrug of l-carnosine (LCS). LCS can prevent cataractogenesis as it scavenges ROS and prevents ROS-induced damage to biomolecules [107]. It has demonstrated a mitigative effect on cataracts in canines [108]. Babizhayev (2004) found that 4 months of use of NACS lubricant eyedrops significantly improved visual acuity and glare sensitivity in 65 older adults with cataracts [109]. Additionally, there were no systemic/ocular adverse effects of the treatment, indicating its tolerability.

While it improves visual outcomes in cataract patients, there is a lack of human trials indicating that NACS eyedrops have a reversal effect on cataracts. Further clinical studies are necessary to consolidate NACS’ potential role in current cataract treatments.

#### 4.2.2. N-acetylcysteine Amide

NAC is a prodrug of a GSH precursor, which is an antioxidant known to mitigate cataractogenesis [118]. Because NAC is negatively charged at physiological pH, it is difficult for it to cross the cell membrane. NACA is an analog of NAC that, due to its neutral amide group, has higher lipophilicity and thus better cell permeability than NAC [111]. Due to this, it is a more powerful antioxidant than NAC. Additionally, it can inhibit cataractogenesis to a greater degree than NAC [110].

Maddirala et al. (2017) tested the effect of NACA on cataract severity in rat pups [111]. They found that intraperitoneal injection of NACA mitigated sodium selenite (Na_2_SeO_3_) induced cataract formation, with pups injected with NACA exhibiting a lower initial severity of cataract formation than pups who received a phosphate buffer control injection. Furthermore, NACA eye drops were sufficient to induce a reversal in cataract grade, with the treatment decreasing the severity of cataracts in the eye drop-treated pups over time. Additionally, the NACA-treated group exhibited significantly increased GSH levels, and significantly lower malondialdehyde (MDA, an oxidative stress marker) and calcium levels which were reduced to non-sodium selenite-treated control levels, illustrating its ability to reduce oxidative stress in the lens. More recent work by Martis et al. (2023) has fortified and expanded upon these findings [110]. In pig and rat lenses, they found that pretreatment with NACA, and another NAC analog called diNACA, was able to significantly reduce lens opacity induced by H_2_O_2_ exposure. NACA exhibited these anti-cataract properties via increasing cysteine and GSH levels in the lens, but diNACA seemed to utilize a different antioxidant mechanism that was unknown to the authors, warranting further research into this pharmacological agent. As such, the use of NACA and diNACA in cataract treatment does have a strong preclinical basis which may warrant clinical investigations in the future. In particular, the ability of NACA to reverse cataract grade as demonstrated by Maddirala et al. is a motivating factor for further investigation, due to the significant potential upside of the development of a medication to reverse cataract grade.

#### 4.2.3. Resveratrol

Resveratrol is a polyphenolic phytoalexin produced in plants that acts as a radical scavenging agent. In vitro evidence suggests that resveratrol’s anti-cataract and anti-oxidant effects are due to the upregulation of endogenous antioxidant enzymes such as catalase, superoxide dismutase-1, and heme oxygenase-1 [119]. Additionally, from a diabetic cataract perspective, it has been demonstrated that resveratrol can reduce oxidative damage from high glucose through the activation of autophagy via the p38 mitogen-activated protein kinase signaling pathway [114]. Its anti-cataract properties have also been studied in animal models of diabetic and naphthalene-induced cataracts. In a rat model of diabetic cataracts, while it did not prevent the eventual formation of diabetic cataracts, supplementation with resveratrol significantly delayed cataract progression compared to control [112]. Additionally, resveratrol caused a decrease in protein carbonyl levels, which is a proxy for oxygen-mediated protein oxidation. This suggests that the effect of resveratrol could be credited at least in part to preventing oxidative damage to proteins in the lens. These findings are also consolidated in other literature. Doses of 20 and 40 mg/kg/day of resveratrol were able to significantly delay lenticular opacity, increase antioxidant levels, and decrease lipid peroxidation in rats with naphthalene-induced cataracts [113].

#### 4.2.4. Baicalein

Baicalein is a flavonoid with known antioxidant, antiapoptotic, and anti-inflammatory functions [116]. It is an aglycone of baicalin, and both are ROS scavengers [120]. Hu et al. (2024) conducted the first study on the pharmacological impact of baicalein for cataracts [116]. They found that baicalein was able to significantly attenuate the dense opacity of the lens and increase the lens’ soluble protein content in juvenile rats with sodium selenite-induced cataracts. In terms of antioxidant effects, they noted that it reduced serum and lens MDA levels and attenuated lens epithelial cell damage, implying an anti-cataract effect. Additionally, it increased superoxide dismutase and glutathione peroxidase levels, which can neutralize ROS. As this is the first pharmacological study of baicalein in the context of cataracts, however, further investigation is pertinent to consolidate these results.

#### 4.2.5. Metformin

Metformin is an anti-aging drug which is used in the treatment of diabetes. It can reduce damage from oxidative stress and therefore inhibit age-related changes [121]. Chen et al. (2022) assessed the efficacy of metformin in combatting senescence of LECs, which contributes to age-related cataracts [117]. They found that treating mice with a chronic low dose of metformin inhibited lens opacity and LEC senescence in mice, and enhanced autophagy, confirming its anti-aging effects.

While current research centered on the anti-cataract effects of NACA, resveratrol, metformin, and baicalein preclinically, it is important to note that chemically induced cataract models strongly resemble human senile cataract development, and thus act as a guide for future clinical applications of these pharmacological agents [122]. Overall, with NACA’s ability to prevent and reverse the established cataract severity, it holds a strong potential for intriguing advances in cataract treatment. Particularly, its ability to reverse cataract grade as exhibited by Maddirala et al. (2017) suggests a role as a potential alternative to cataract surgery [111]. Similarly, the roles of resveratrol and baicalein as antioxidant attenuators of oxidative damage to lens components, in conjunction with their antioxidant effects, present other intriguing avenues for future non-surgical anti-cataract treatment.

### 4.3. Nanotechnology-Based Drug Delivery Systems for Cataract Prevention and Treatment

Nanotechnology-based drug delivery systems have been developed to control and enhance the longevity of drug release. In terms of treatments for cataracts, the majority of nanotechnology-based products work to improve pre-existing clinical and preclinical treatments (Table 2).

#### 4.3.1. N-acetylcarnosine Nanoparticles

Wang et al. (2018) conducted the encapsulation of NACS into gold nanoparticles, intending to innovate in this therapy option for cataracts [123]. They investigated the cytotoxicity of the bio-fabricated nanoparticles in vitro and found that NACS exhibited toxicity at high concentrations (80 and 100 ppm) in fibroblast cells, but this was significantly reduced by encapsulation in gold particles. Thus, the ability of this encapsulation method to enhance NACS’ biocompatibility helps improve current NACS-based therapies. However, these nanoparticles have not been tested directly for their ability to prevent cataract formation and progression, which is a limitation of current research on this mode of treatment.

#### 4.3.2. Resveratrol Nanoparticles and Nanovesicles

Vora et al. (2019) developed novel lipid-cyclodextrin-based nanoparticles loaded with resveratrol to improve its poor bioavailability [124]. They tested the antioxidant ability of their nanoparticles by measuring GSH and superoxide dismutase activity in cow lens cultures. They found that the levels of both markers were significantly higher when they utilized their resveratrol-loaded nanoparticles as opposed to resveratrol alone, indicating that these particles could act as a better delivery system for resveratrol in vivo. Machado et al. (2021) developed niosomes encapsulating resveratrol to address its chemical instability, as it undergoes *trans*-*cis* isomer conversion via light irradiation [115]. They found that their niosomes were able to maintain the antioxidant capacity of the RSV while also preserving approximately 85% of *trans*-resveratrol, which is the more bioactive isomer. Each of these studies demonstrate promising nanotechnology-based delivery systems of resveratrol that can be considered for use in clinical studies of the drug.

#### 4.3.3. Baicalin

Li et al. (2022) developed chitosan-coated mPEG-PLGA nanoparticles (BA LCH NPs) to deliver baicalin [125]. Baicalein, which was mentioned in the previous section, is an aglycone of baicalin, and both are ROS scavengers that have been tested for cataract treatment [120]. They found that BA LCH NPs had a higher cellular uptake compared to baicalin. In vivo, BA LCH NPs also had higher corneal retention in rabbits with selenite-induced cataracts. They raised levels of catalase, superoxide dismutase, and glutathione peroxidase, while decreasing MDA levels, indicating an antioxidant effect. This trend was stronger with BA LCH NPs than it was with baicalin alone. Overall, this work provides a drug delivery system that can improve potential baicalin-based anti-cataract treatments by enhancing its bioavailability.

#### 4.3.4. Cerium Oxide

Zhou et al. (2019) developed cerium oxide (CeO_2_) nanoparticles coated in PEG-PLGA as a potential treatment for diabetic cataracts [126]. CeO_2_ reduces neuroinflammation and cellular ROS levels by catalyzing reactions with superoxide and H_2_O_2_ [127,128]. PEG and PLGA are biodegradable and biocompatible polymers, which help the nanoparticles’ solubility and biocompatibility [126]. They found that the addition of these nanoparticles significantly decreased peroxide and superoxide concentrations in an in vitro oxidative injury model using lens epithelial HLE-B3 cell cultures. They also found that subcutaneous injection of a high concentration of nanoparticles in diabetic rats in vivo significantly reduced lens MDA levels, significantly increased lens glutathione levels, and significantly attenuated diabetic cataract development with no obvious immune reactions [126]. Cerium oxide is a widely researched molecule for its biomedical applications and antioxidant properties [128]. This study provides a pharmaceutical formulation of CeO_2_ that is water-soluble through its PEG-PLA coating, which has been a challenge in the clinical application of this molecule [128]. However, it is limited in that the mechanism by which the antioxidant effects are observed in the lens is still unclear. Regardless, CeO_2_ nanoparticles are strong candidates for further research in cataract treatment. Additionally, due to their interference with ROS, they are strong candidates for AMD treatment as well.

### 4.4. Gene Therapy for Cataract Prevention and Treatment

Several gene therapy options have been investigated to treat cataracts, particularly in the area of preventing posterior capsule opacification, which is the leading complication of cataract surgery resulting in visual impairment with an incidence of 30% [129,130]. Posterior capsule opacification (PCO) results from a combination of many processes, including the proliferation of LECs onto the posterior lens capsule following cataract surgery [131]. Additionally, LECs undergo epithelial–mesenchymal transition (EMT), which is another major factor in the pathology of this complication [131]. Oxidative stress has been implicated EMT of human LECs, as it can induce cytokine expression, which has been positively associated with EMT markers, suggesting a role in PCO development [132].

Currently, YAG laser is the leading treatment for PCO, but this laser treatment has complications such as retinal detachment and macular edema [130]. Additionally, it does not work to prevent PCO, but instead to treat it. As such, gene therapy has been considered as a means to prevent the action of factors involved in PCO pathology and thus attenuate PCO incidence (Table 3).

#### 4.4.1. Suicide Gene Therapy

Suicide gene therapy has been considered for PCO prevention. It is a strategy in which transgenes inducing cell death in HLECs are introduced into the proliferating and migrating cells following cataract surgery [133,134,140,141,142]. Malecaze et al. (2005) overexpressed various apoptotic molecules in rabbit lenses [133]. They found that one in vivo injection of procaspase 3 or Bax delivered to the capsular bag following phacoemulsification in an adenovirus vehicle was sufficient to prevent PCO in rabbits. Following this, Malecaze et al. (2006) described a method to over-express these pro-apoptotic molecules in LECs to prevent PCO [134]. They spatially and genetically restricted this gene transfer in rabbit residual LECs. They found that the use of human serotype 5 adenovirus expressing LEP503, MIP, and Filensin promoters induced strong expression of the reporter gene in HLECs.

In a related study, Jiang et al. (2011) shared an enhanced Cre/LoxP system, in which a lentiviral vector was utilized to express Cre in human LECs using LEP503, a lens-specific promoter, to express the genes in HLECs [135]. Specifically, herpes simplex virus thymidine kinase HSV-tk (HSV-tk) was expressed in the cells, which were then treated with ganciclovir (GNV) in order to induce cell death. They found that this method of gene therapy was effective in targeting HLECs as opposed to other ocular cell types and preventing their in vitro proliferation.

In combination, these results suggest a potential of suicide gene therapy for PCO prevention, and importantly, introduce a method to genetically restrict this therapy to HLECs. This is important to reduce any off-target effects of gene therapies, which is especially critical with suicide gene therapy.

#### 4.4.2. RNA Interference

RNA interference (RNAi) is a gene therapy method by which small strands of RNA prevent protein translation through the binding of mRNAs coding for target proteins. With regards to cataracts, RNAi has been employed to assess targets for PCO prevention.

Epidermal growth factor (EGF) is a major factor involved in cell proliferation. Huang et al. (2011) tested the effect of EGF receptor (EGFR) small interfering RNA (siRNA) on HLECs to induce a knockdown [136]. They tested the effects of this interference on HLEC cultures and found that it was sufficient to inhibit cell proliferation and block the EGF–EGFR signal pathway. Furthermore, they tested this siRNA in a rat PCO model and found that it significantly reduced PCO extent.

Moving on to EMT, discoidin I-like domain-containing protein 3 (EDIL3) is an extracellular matrix protein known to participate in EMT of some cell types. Zhang et al. (2018) found that, in HLECs, in vitro RNAi-mediated knockdown of *EDIL3* was able to significantly reduce both HLEC proliferation and migration [137]. They therefore hypothesized that this molecule could be involved in PCO pathogenesis, making it a target for gene therapy.

In addition to the aforementioned targets, some RNAi therapies have been tested either in vivo or in vitro to prevent TGF-βRII-induced EMT. TGF-βRII is a receptor that binds TGF-βII, resulting in downstream signaling related to EMT of LECs. Zheng et al. (2012) tested siRNAs targeting TGF-βRII in cultured LECs [138]. They found that RNAi significantly attenuated LEC migration through wound scratch assays, providing evidence for its role in EMT. Building on this, *Snail* is a transcription factor which triggers TGF-βII-activated EMT. Li et al. (2013) utilized RNAi to target *Snail* using siRNA [139]. They found that in a culture of the human epithelial cell line HLEB3, *Snail* siRNA caused significant inhibition of TGF-βII-activated EMT and *Snail* expression. Finally, integrin-linked kinase (ILK) is another molecule involved in EMT and is specifically part of the TGF-β pathway. Zheng et al. (2016) tested ILK RNAi using short hairpin RNA (shRNA) delivered via a lentiviral vector [143]. They found that cells transfected with ILK shRNA exhibited significantly less migration, higher apoptosis, and arresting of the cell cycle at the G1/S transition.

#### 4.4.3. CRISPR-Cas9

The CRISPR/Cas9 is a genome editing technique in which an endonuclease cleaves DNA at a site specified by guide RNA (RNA), permitting genetic modification [144]. In the first study on gene-editing for PCO treatment, Wang et al. (2024) sought to knock out TGF-βRII using the CRISPR/Cas9 system via in vitro and in vivo lentiviral transfection of LECs [130]. For the in vivo experimentation, rabbits received an intravitreal injection of rAAV virus following phacoemulsification surgery and intraocular lens implantation. Cell proliferation was assessed via HLE-B3 culture. They found that intravitreal injection of the vector significantly suppressed PCO and in vitro transfection of HLECs significantly decreased their proliferation, suggesting a role for CRISPR-Cas9 in PCO prevention.

## 5. Challenges and Limitations of Antioxidants and Novel Therapeutic Approaches

The current gold standard for cataract treatment is surgical removal and lens replacement. However, the cost of surgery and its various complications, such as posterior capsule rupture, endophthalmitis, and the need for additional surgery, indicate a demand for alternative preventative and treatment measures for this condition [105]. While epidemiological evidence suggests a role for dietary supplements in cataract prevention, clinical studies do not support an appreciable effect of vitamin C, vitamin E, lutein, and zeaxanthin for cataract treatment or prevention. Alternatively, promising antioxidant pharmacological treatments do exist that can prevent and reverse cataracts, which have been enhanced with nanotechnology. However, these treatments, along with gene therapy options, are largely preclinical, and thus require further clinical evidence to consolidate a role in cataract treatment.

Finally, there are many avenues that are currently underexplored for gene therapy treatments. Genetic mutations are responsible for close to a third of congenital cataracts, and researchers have identified gene repair targets to address this [145]. For example, GJA8, GJA1, and GJA3, encoding connexin 50, 43, and 46, respectively, have been identified as vital in preventing cataract formation [145,146]. In addition to gene repair targets, other potential targets exist for silencing via gene therapy to attenuate lens opacification. For example, connective tissue growth factor has a demonstrated involvement in the proliferation, EMT, and migration of LECs [147]. Furthermore, circular DNA gene silencing targets involved in LEC proliferation, such as Hsa_circRNA_0060640, have also been identified [148]. As such, there exist many gene therapy targets that are currently underexplored.

## 6. Recommendations and Future Directions

Although the importance of agents promoting an antioxidative environment in the prevention of cataractogenesis has been underlined within this comprehensive literature review, recommendations must be critically interpreted. The majority of the studies suggesting a beneficial role of dietary nutrients (i.e., vitamins C and E, lutein, and zeaxanthin) in the prevention of cataract formation are epidemiological studies and meta-analyses, therefore suggesting a possible association. In AREDS2, a robust double-masked clinical trial, the daily consumption of supplementary lutein and zeaxanthin was not shown to affect vision loss and rate and cataract surgery incidence in participants, therefore suggesting no additional benefits in lutein and zeaxanthin dietary intake. However, in this study, lutein and zeaxanthin were used at doses of 10 mg and 2 mg, respectively. Therefore, further controlled clinical trials aiming to assess the dosage-dependent effect of dietary nutrients with antioxidative capacities on cataract formation are required. Preclinical studies regarding the use of NACS, NACA, resveratrol, and baicalein are in favour of positive antioxidative effects on cataract progression, suggesting weak recommendations for their use in cataract prevention. Overall, the current state of knowledge cannot provide strong recommendations on the use of antioxidants for the prevention of cataract formation. Certain agents, such as N-acetylcarnosine, can be suggested in patients with positive cataract-related symptoms (e.g., glare) who are uncertain with surgical treatment as an alternative. However, clinicians should be aware of the lack of evidence-based studies for such applications.

## 7. Conclusions

Cataracts are a leading cause of vision impairment worldwide. A pivotal mechanism involved in cataractogenesis is the dysregulation in antioxidative regulatory mechanisms and cell proliferation and cell death pathways within the lens epithelium. These abnormal cell-signaling pathways lead to protein aggregates, cross-linking, and light-scattering phenomena in cataract formations. Numerous efforts have been deployed over the past years to investigate the role of antioxidants in cataract formation prevention and treatment. Observational studies have suggested a potential association between the use of antioxidants and delaying cataract formation. Preclinical studies have demonstrated the importance of antioxidants on PCO and lens epithelium proliferation. Emerging technologies, such as gene therapy, have further shed light on possible non-surgical therapeutic avenues. Overall, the rapidly evolving field provides hope for future non-surgical therapeutic treatments for cataracts and possible interventions in age-specific patients to delay cataract formation. 

## Figures and Tables

**Figure 1 biomolecules-14-01055-f001:**
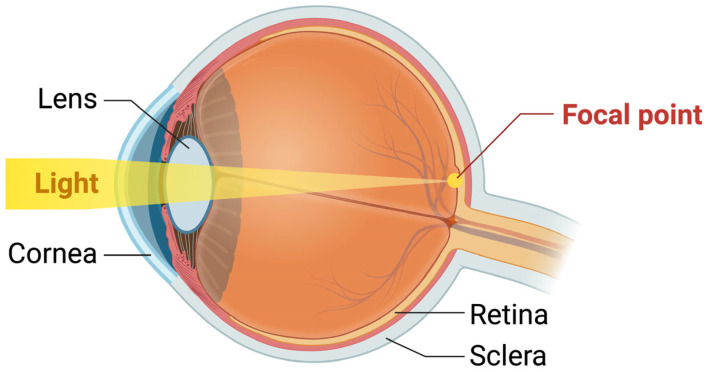
Optics in ophthalmology: image refraction. As light passes through the structures of the human eye (cornea, lens, vitreous humor), it is refracted until it reaches the focal point situated at the back of the posterior segment. Each anatomical structure within the eye possesses a refractive index, which is a measure of the bending of a ray. Structural alterations to these structures cause a change in their respective refractive index, therefore leading to vision impairment. The Figure was created with BioRender.com (https://www.biorender.com, accessed on 03 July 2024).

**Figure 2 biomolecules-14-01055-f002:**
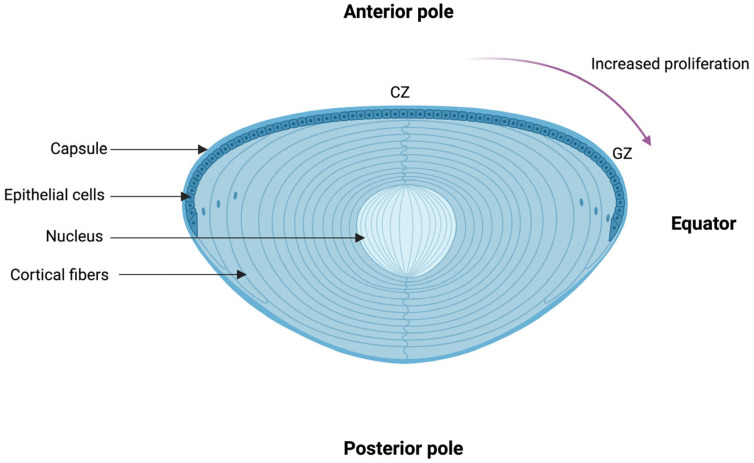
Schematic illustration of the human eye lens. The human eye lens is composed of four main structures: the capsule, epithelial cells the nucleus, and the cortical fibers. As the epithelial cells migrate from the anterior pole to the posterior pole, they acquire greater proliferative capacity; in the central zone (CZ), there is an absence of mitosis in epithelial cells whereas, in the germinal zone (GZ), the epithelial cells exhibit maximal mitosis and cell proliferation. Alterations in the proliferation and migration of epithelial cells form the backbone of cataracto-genesis. The Figure was created with BioRender.com (https://www.biorender.com, accessed on 05 July 2024.

**Figure 3 biomolecules-14-01055-f003:**
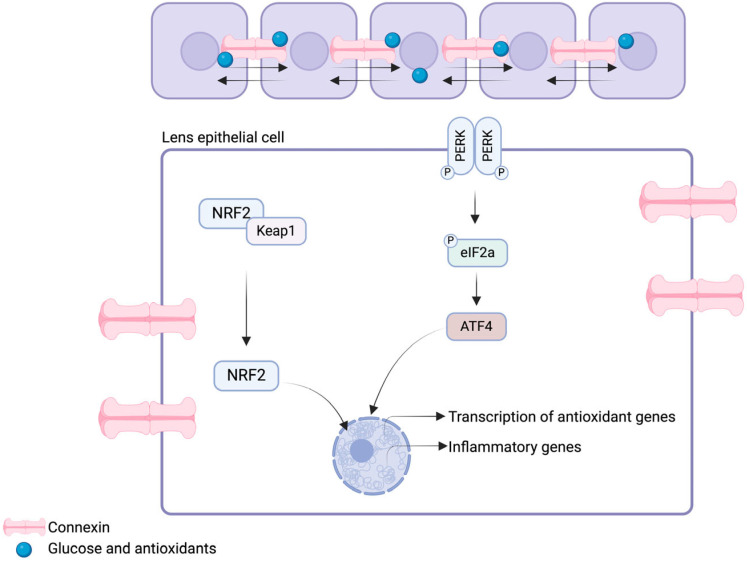
Schematic representation of antioxidant mechanisms in the human lens epithelium. Antioxidants and glucose are delivered to lens epithelium cells through the gap junctions known as connexins. Upon oxidative stress in the human eye lens, NRF2 and ATF4 are activated to induce the transcription of antioxidant and inflammatory genes. The figure was created with BioRender.com (https://www.biorender.com, accessed on 10 July 2024).

**Table 1 biomolecules-14-01055-t001:** Summary of antioxidative pharmacological agents for the treatment of cataracts.

Treatment	Structure and Description	Implications for Cataract Treatment	References
N-acetylcarnosine	Prodrug of l-carnosine. 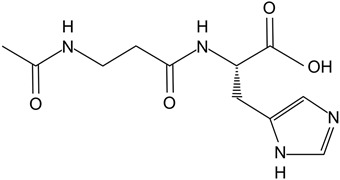	Reduces lens opacification in canine cataracts, NACS eyedrops improve visual acuity and glare sensitivity in humans with cataracts.	[107,108,109]
N-acetylcysteine amide	Analog of NAC, a glutathione prodrug. 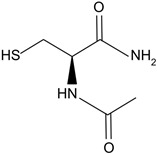	NACA intraperitoneal injection prevents sodium selenite-induced cataract formation in rats, NACA eye drops reverse sodium selenite-induced cataract grade in rats, NACA and diNACA reduce H_2_O_2_-induced lens opacity in pig and rat lenses, with NACA increasing antioxidant levels as well.	[110,111]
diNACA	Analog of NACA. 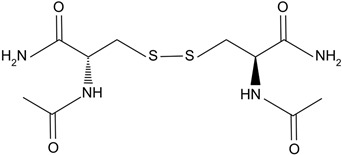		
Resveratrol	Polyphenolic phytoalexin produced in plants, *trans* isomer is more bioactive. 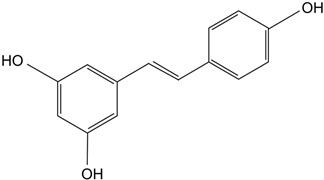	Delays diabetic cataract formation in rats, mitigates oxygen-mediated protein oxidation in diabetic rats, protects human lens epithelial cells against oxidative damage, increases antioxidant levels and delays lenticular opacity in rats with naphthalene-induced cataracts.	[112,113,114,115]
Baicalein	Antioxidant flavonoid. 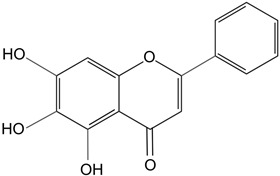	In rats with sodium-selenite induced cataracts, it decreases dense opacity of the lens, increases soluble protein content, reduces oxidative stress, and prevents damage of lens epithelial cells.	[116]
Metformin	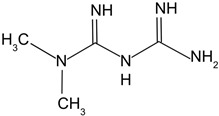	Chronic low dose of metformin in mice significantly decreased lens opacity and lens epithelial cell senescence, which increasing autophagy.	[117]

**Table 2 biomolecules-14-01055-t002:** Summary of most recent nanotechnologies formulated to improve treatment outcomes for cataracts.

Drug	Nanotechnology Used	Outcomes	Reference
NACS	Encapsulated NACS into gold nanoparticles.	Attenuated NACS toxicity at high concentrations, increased biocompatibility and bioavailability.	[123]
Resveratrol	Encapsulated resveratrol into lipid cyclodextrin-based nanoparticles.	Increased levels of antioxidant markers in bovine lens cultures to a higher degree than resveratrol alone.	[124]
Resveratrol	Encapsulated resveratrol into niosomes.	Maintained antioxidant capacity of resveratrol, prevented light irradiation-induced isomer conversion of resveratrol to its less bioactive *cis* isomer.	[115]
Baicalin	Encapsulated baicalin into chitosan-coated mPEG-PLGA nanoparticles.	Increased cellular uptake of baicalin, increased corneal retention of baicalin in rabbits, increased antioxidant levels and decreased oxidative stress markers in rabbits with selenite-induced cataract to a greater degree than baicalin alone.	[125]
CeO_2_	Encapsulated CeO_2_ in PEG-PLGA coated nanoparticles.	Allowed for water soluble formation of CeO_2_ suitable for biological use. Decreased peroxide and superoxide concentrations in lens epithelial cell cultures. Reduced oxidative stress markers, increased antioxidant levels, and attenuated cataract development in rats with diabetic cataracts.	[126]

**Table 3 biomolecules-14-01055-t003:** Summary of gene therapy studies focused on posterior lens opacification prevention.

Gene(s) of Interest	Outcomes	Reference
Suicide Gene Therapy		
Procaspase 3 or Bax	Overexpression of pro-apoptotic molecules was successfully targeted to rabbit residual lens epithelial cells, and sufficiently prevented PCO in rabbits.	[133,134]
HSV-tk (plus treatment with GNV)	HSV-tk was successfully expressed in HLECs and, when treated with GNV, was able to cause cell death.	[135]
RNA Interference		
EGF	siRNA successfully inhibited cell proliferation of HLECs and significantly reduced PCO in a rat model.	[136]
EDIL3	Knockdown significantly reduced HLEC proliferation and migration in vitro.	[137]
TGF-βRII	RNAi significantly reduced LEC migration.	[138]
Snail	siRNA successfully inhibited TGF-βII-mediated EMT of human epithelial cells.	[139]
ILK	shRNA significantly decreased migration, increased apoptosis, and caused arresting of cells at G1/S transition.	[138]
CRISPR-Cas9		
TGF-βRII	TGF-βRII knockout caused significant decrease in PCO incidence for rabbit PCO model, as well as significant decreased in in vitro HLEC proliferation.	[130]

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
