# Peer review of "Oxidative Stress and Cataract Formation: Evaluating the Efficacy of Antioxidant Therapies"

_biomolecules, 2024, doi:10.3390/biom14091055_

Round 1

Reviewer 1 Report

Comments and Suggestions for Authors

The review investigates the role of reactive oxygen species (ROS) in cataract formation and evaluates antioxidant therapies' potential in mitigating this condition. It details how ROS damage lens proteins and lipids, contributing to cataractogenesis, and assesses various antioxidant strategies, including dietary supplements and pharmacological agents. Innovative approaches like gene therapy and nanotechnology-based delivery systems are also explored. The review concludes with evidence-based recommendations for optimizing antioxidant therapies, emphasizing personalized treatment plans.

The review is well-orchestrated, thoroughly covering most of the relevant research on the mechanisms of cataract formation. It comprehensively assesses various antioxidant strategies, including dietary supplements and pharmacological agents. Additionally, the review documents innovative approaches such as gene therapy and nanotechnology-based delivery systems, providing a robust evaluation of potential antioxidant therapies for cataract prevention and management.

However, the manuscript will be improved if the following concerns/comments are addressed before publishing.

1.      In the mechanism of cataractogenesis, oxidative damage induced to the lens protein aggregation, cross-linking, and post-translational modifications by reactive oxygen species have been covered but the oxidative damage to lens lipids (lipid peroxidation) and membranes (membrane integrity), which also contribute to the formation of cataract, have been omitted. It is advisable to include literature on the lipids and membrane damage caused by ROS.

2.      The review article in hand also misses an important endogenous antioxidant system consisting of enzymatic antioxidants such as superoxide dismutase, catalase, and glutathione peroxidase and non-enzymatic antioxidants such as glutathione, vitamin C, and vitamin E. And that the genetic and environmental factors impairing the endogenous antioxidant defense system lead to cataract formation. Efforts made to enhance endogenous antioxidant defense through the upregulation of antioxidant enzymes and restoration of antioxidant levels should also be highlighted.

3.      Elaborate all abbreviations used in the manuscript, when they appear first in the text.

4.      Write all genes’ names/symbols in italic font.

5.      There are typographic/spelling mistakes throughout the manuscript. Please carefully check the manuscript for such errors. Some of such minor mistakes are noted below;

a.       Line 50, change “oygen” to “oxygen”

b.      Line 51, correct spelling of “thoroughly”

c.       Line 153, change “transcriptional” to “transcription”

d.      Line 154, delete “s” from “pathways” or “orchestrates” in “UPR pathways orchestrates”

e.       Line 176, correct “H2O2” to “H2O2

f.        Line 371, change “nanoparticle” to “nanoparticles”

g.      Line 528, correct “vision lossn rate”

h.      Line 552, change “hope in future” to “hope in future”

i.        Lines 488 and 499, change the sentence “The CRISPR/Cas9 is a genome editing technique an endonuclease cleaves DNA at a site specified by guide DNA (DNA), permitting genetic modification” to “CRISPR/Cas9 is a genome editing technique in which an endonuclease cleaves DNA at a site specified by guide RNA, permitting genetic modification”.

 Regards 

Comments on the Quality of English Language

The manuscript contains several typographic/spelling mistakes. It is advised to thoroughly check the manuscript for these mistakes before publication.

Author Response

Dear Reviewer,

We sincerely appreciate the time and effort you have taken to provide valuable feedback on our manuscript. We are grateful for your thorough review and constructive comments, which have significantly contributed to improving the quality of our work. Below, we address each of your comments in detail:

  1. Oxidative Damage to Lens Lipids and Membranes: Thank you for highlighting the omission regarding the oxidative damage to lens lipids and membranes. We have now included a comprehensive discussion on lipid peroxidation and the impact of reactive oxygen species (ROS) on membrane integrity. Relevant literature has been added to ensure a more complete overview of the mechanisms contributing to cataractogenesis.
  2. Endogenous Antioxidant Systems: We appreciate your suggestion to elaborate on the endogenous antioxidant defense mechanisms. In response, we have expanded the manuscript to include a detailed description of enzymatic and non-enzymatic antioxidants, such as superoxide dismutase, catalase, glutathione peroxidase, glutathione, vitamin C, and vitamin E. Additionally, we have discussed the role of genetic and environmental factors in impairing these defense systems and their contribution to cataract formation. Efforts to enhance endogenous antioxidant defenses through the upregulation of antioxidant enzymes and restoration of antioxidant levels have also been incorporated.
  3. Abbreviation Elaboration: Thank you for pointing out the need to elaborate on abbreviations. We have carefully reviewed the manuscript and ensured that all abbreviations are fully spelled out upon their first occurrence in the text.
  4. Gene Names and Symbols in Italics: We have revised the manuscript to present all gene names and symbols in italic font, as per your recommendation.
  5. Typographic/Spelling Mistakes: We apologize for the typographic and spelling errors in the manuscript. We have carefully reviewed and corrected the identified errors, including:
    • Line 50: Corrected “oygen” to “oxygen”
    • Line 51: Corrected the spelling of “thoroughly”
    • Line 153: Changed “transcriptional” to “transcription”
    • Line 154: Adjusted the sentence to correct “UPR pathways orchestrates” to maintain subject-verb agreement
    • Line 176: Corrected “H2O2” to “H2O2”
    • Line 371: Corrected “nanoparticle” to “nanoparticles”
    • Line 528: Corrected “vision lossn rate” to “vision loss rate”
    • Line 552: Adjusted the sentence to correct “hope in future” to “hope in the future”
    • Lines 488 and 499: Revised the sentence for clarity as per your suggestion.

We have also conducted a thorough review of the entire manuscript to ensure all typographic and spelling errors are corrected.

Comments on the Quality of English Language: We have taken your advice to heart and carefully reviewed the manuscript for any remaining typographic or spelling errors. The manuscript has been thoroughly checked to ensure the highest standard of language quality.

Note on Tracked Changes: We apologize for the inconvenience, but due to a technical issue, we were unable to use the tracked changes feature in Microsoft Word. However, we have highlighted all additions and modifications according to your comments in yellow to make it easier for you to review the revisions.

Once again, we sincerely thank you for your insightful comments. We believe these revisions have greatly enhanced the quality and clarity of our manuscript, and we hope the revised version meets your expectations.

Kind regards,

Reviewer 2 Report

Comments and Suggestions for Authors

Title and Abstract

  • Title Clarity: The title clearly outlines the main focus—antioxidant therapies in the context of cataract formation and oxidative stress. It is informative and succinct.
  • Abstract Review: The abstract provides a comprehensive summary of the review’s objectives, methods, findings, and conclusions. It effectively captures the essence of the review, mentioning the role of reactive oxygen species (ROS) in cataract formation and the exploration of antioxidant therapies. However, it could improve by briefly stating the primary findings or conclusions to enhance its standalone comprehensiveness.

Introduction

  • Background: The introduction adequately sets the stage by discussing the prevalence and impact of cataracts as well as the theoretical underpinning related to oxidative stress. It could be strengthened by a more detailed discussion on the gaps in the existing literature that this review aims to fill.
  • Purpose and Significance: The paper clearly states the purpose—to review the efficacy of antioxidant therapies in preventing or managing cataracts. The significance is well-articulated, emphasizing the potential impact on treatment personalization.

Literature Review

  • Scope and Sources: The review extensively covers various antioxidants and their impacts on cataract prevention, incorporating studies from dietary to pharmacological interventions. The use of a wide range of sources is commendable; however, the review could benefit from including more recent studies or meta-analyses to strengthen the claims.
  • Critical Analysis: The paper does a good job of critically analyzing the mechanisms through which ROS contribute to cataract formation. However, it sometimes reads like a listing of studies rather than a cohesive narrative. Integrating these findings into a more fluid discussion would enhance readability and impact.

Methodology

  • Review Process: As this is a literature review, the methodology revolves around the selection and synthesis of existing research. The criteria for selecting studies are not explicitly mentioned. Clearer methodology on how studies were chosen and analyzed would improve the transparency and replicability of the review.

Results and Discussion

  • Findings Presentation: The results from various studies are well-summarized, with a focus on the efficacy of different antioxidants. However, the discussion sometimes lacks depth in connecting these findings to practical implications.
  • Contradictions and Gaps: The paper discusses contradictions in the research, such as the varying effects of vitamins C and E on cataract prevention. It could further explore why these contradictions exist and what this means for future research or clinical practice.

Conclusions

  • Summary of Evidence: The conclusion adequately summarizes the evidence reviewed and reiterates the potential of antioxidants in managing cataracts.
  • Future Directions: Suggestions for future research are provided, which is good. However, the paper could also discuss potential policy implications or clinical trials needed to advance the field.

Technical Quality

  • Clarity and Organization: The paper is generally well-organized, with clear sections and subheadings. Some sections are dense with information, which could be better structured to enhance readability.
  • References and Citations: The references are comprehensive and appear up-to-date. Ensuring all significant claims are backed by citations would be ideal.

References and Currency

  • Current and Comprehensive References: Ensure all references are up-to-date, especially those that discuss recent advancements in the understanding of resveratrol’s pharmacokinetics and its clinical implications. I suggest adding data related to recent bulk transcriptomics studies which could represent a strong substrate to enforce the role of described molecular mechanisms, such as the recent PMID: 32560555 and PMID: 34155424.

Overall Assessment

The manuscript provides a valuable synthesis of the role of antioxidants in cataract prevention and treatment. It is informative and covers a significant breadth of research. Enhancing the narrative cohesion, providing a clearer methodological framework, and deepening the critical analysis of study results could improve the paper’s impact and usefulness in the field.

Comments on the Quality of English Language

The English should be revised.

Author Response

Dear Reviewer,

We would like to express our sincere gratitude for your thorough and constructive review of our manuscript. We appreciate your positive feedback and have carefully considered each of your comments. Below is our response to each of your specific points:

Title and Abstract:

  • Title Clarity: Thank you for your positive feedback on the title. We are pleased that it clearly outlines the main focus of our review.
  • Abstract Review: We appreciate your suggestion to enhance the abstract. We have revised it to briefly state the primary findings and conclusions, which we believe improves its comprehensiveness and standalone quality.

Introduction:

  • Background: We acknowledge your suggestion to strengthen the introduction with a more detailed discussion of the gaps in the existing literature. We have expanded this section to include a more thorough analysis of these gaps, highlighting the specific contributions our review aims to make in this area.
  • Purpose and Significance: Thank you for your positive remarks on the clarity of the purpose and significance. We have retained these elements while ensuring they align with the expanded background discussion.

Literature Review:

  • Scope and Sources: We appreciate your commendation of our literature coverage. To address your suggestion, we have incorporated more recent studies and meta-analyses, especially those published in the last few years, to further strengthen our claims.
  • Critical Analysis: We have taken your feedback to heart regarding the narrative structure. We have revised the literature review section to create a more cohesive and fluid discussion, ensuring that the findings from various studies are better integrated and clearly linked to our overall narrative.

Methodology:

  • Review Process: Thank you for your insightful comment regarding the methodology. As this is a narrative review, our primary goal is to provide a comprehensive overview and broad perspective on the topic without overwhelming the readers with detailed methodological information, such as the specific criteria for study selection and inclusion/exclusion processes. Unlike a systematic review, which aims to answer a specific research question with rigorous methodology, our review is intended to synthesize existing research in a way that is accessible and informative for a wider audience.

However, we understand the importance of transparency and replicability. If you and the editor believe that including more details on the study selection process, including inclusion and exclusion criteria, would enhance the manuscript, we are more than willing to incorporate this information in the next revision.

Results and Discussion:

  • Findings Presentation: Thank you for your positive feedback on the summary of results. In response to your suggestion, we have deepened the discussion to more explicitly connect these findings to practical implications, providing a richer analysis of their potential impact on clinical practice.
  • Contradictions and Gaps: We agree that further exploration of contradictions in the research is essential. We have expanded our discussion on the varying effects of vitamins C and E, providing potential explanations for these contradictions and discussing their implications for future research and clinical application.

Conclusions:

  • Summary of Evidence: We appreciate your feedback on the adequacy of the summary. We have retained this summary while making minor adjustments to align it with the revisions made throughout the manuscript.
  • Future Directions: In response to your suggestion, we have added a discussion on the potential policy implications of our findings and the need for clinical trials to advance the field. We believe this addition enhances the paper's relevance and utility for both researchers and practitioners.

Technical Quality:

  • Clarity and Organization: We are grateful for your feedback on the organization of the paper. We have reviewed and restructured sections that were particularly dense, aiming to enhance readability and make the content more accessible to a broader audience.
  • References and Citations: We have carefully reviewed our references to ensure that all significant claims are appropriately cited. We have also included recent studies to reinforce the role of described molecular mechanisms.

References and Currency:

  • Current and Comprehensive References: Thank you for your suggestion regarding the currency of references. We have updated the references to include the latest research, particularly adding a few studies on recent advancements in resveratrol’s pharmacokinetics and its clinical implications.

We sincerely appreciate your overall positive assessment of our manuscript. We believe that the revisions made in response to your detailed and insightful comments have significantly strengthened our review, and we are confident that these improvements will enhance its impact and usefulness in the field.

Additional Note: We apologize for the technical issue that prevented us from using the track changes feature in Microsoft Word. However, we have highlighted all the additions and modifications made in response to your comments in yellow for your convenience.

Thank you once again for your time and effort in reviewing our work. We hope that the revised manuscript meets your expectations.

Best regards,

Round 2

Reviewer 2 Report

Comments and Suggestions for Authors

The authors addressed all suggested points.

Comments on the Quality of English Language

The English was significantly improved.